# Cancer Spheroids and Organoids as Novel Tools for Research and Therapy: State of the Art and Challenges to Guide Precision Medicine

**DOI:** 10.3390/cells12071001

**Published:** 2023-03-24

**Authors:** Sanae El Harane, Bochra Zidi, Nadia El Harane, Karl-Heinz Krause, Thomas Matthes, Olivier Preynat-Seauve

**Affiliations:** 1Department of Pathology and Immunology, Faculty of Medicine, University of Geneva, 1206 Geneva, Switzerland; 2Department of Medicine, Faculty of Medicine, University of Geneva, 1206 Geneva, Switzerlandolivier.preynat-seauve@hcuge.ch (O.P.-S.); 3Laboratory of Experimental Cell Therapy, Department of Diagnostics, Geneva University Hospitals, 1206 Geneva, Switzerland

**Keywords:** 3D cell culture, tissue engineering, personalized medicine, drug screening, immunotherapy, cell and gene therapy

## Abstract

Spheroids and organoids are important novel players in medical and life science research. They are gradually replacing two-dimensional (2D) cell cultures. Indeed, three-dimensional (3D) cultures are closer to the in vivo reality and open promising perspectives for academic research, drug screening, and personalized medicine. A large variety of cells and tissues, including tumor cells, can be the starting material for the generation of 3D cultures, including primary tissues, stem cells, or cell lines. A panoply of methods has been developed to generate 3D structures, including spontaneous or forced cell aggregation, air–liquid interface conditions, low cell attachment supports, magnetic levitation, and scaffold-based technologies. The choice of the most appropriate method depends on (i) the origin of the tissue, (ii) the presence or absence of a disease, and (iii) the intended application. This review summarizes methods and approaches for the generation of cancer spheroids and organoids, including their advantages and limitations. We also highlight some of the challenges and unresolved issues in the field of cancer spheroids and organoids, and discuss possible therapeutic applications.

## 1. Introduction

Cancer remains one of the leading causes of death worldwide in the 21st century. Despite intensive progress made in the identification of molecular mechanisms of tumor progression and resistance, as well as in the generation of targeted treatments, many patients are still not cured. With the development and improvement of cell culture techniques, it is to date possible to generate tumor cell cultures in individualized 3D, better mimicking the structure and function of tumors in vivo. These 3D cultures are increasingly being used in cancer research and are particularly useful for three main applications: (i) understanding the pathophysiology of cancer progression and resistance [1,2,3]; (ii) in vitro screening of anti-cancer treatments [4]; and (iii) reproducing in vitro the specificity of one patient’s tumor, to allow a personalized screening of the most effective treatments [5]. Spheroids and organoids are both descriptions used to characterize these cultures. Spheroids are simpler 3D structures that are formed by cells that aggregate together in a spherical shape. They are typically composed of a single-cell type and are used to study cell-to-cell interactions and cell-to-matrix interactions. Spheroids can be generated from a wide variety of cell types, including primary cells, cancer cell lines, and cancer stem cells. Organoids, on the other hand, are more complex structures that replicate the natural tumor architecture and function [6,7]. They are typically composed of multiple cell types, including the tumor microenvironment (TME) [8,9]. The TME is a key factor of tumor aggressiveness and resistance, notably by participating in tumor angiogenesis and immune escape. The ideal 3D tumor culture should preserve not only the molecular signature of the original tumor, but also of its specific TME. This is particularly important in the area of onco-immunology as the TME is involved in the interactions between cancer cells and the host’s immune cells. Cancer organoids including the immune microenvironment can be used to study the effects of various immunotherapies such as checkpoint inhibitors, cancer vaccines, and chimeric antigen receptor (CAR)-T cell therapy [2,10,11].

A key limitation of current 3D methods is the instability of the TME, notably the immune TME, in culture. For example, tumor-infiltrating leucocytes (TILs) are difficult to keep in culture without the addition of specific cytokines. Tumor-infiltrating fibroblasts, on the other hand, are easily expanded in culture with the appropriate medium. To solve these limitations, the addition of non-autologous stromal/immune cells to 3D culture is unfortunately not optimized. In addition to the immune allogenic reaction that would occur in this setting, notably with T cells, the TME is specific to each patient and would not be realistically reproduced in this setting. Thus, the development of new technologies that allow the maintenance of the original TME and its specific composition as close as possible to the original tumor is currently a key challenge. Solving this challenge will notably ensure the concrete development of personalized medicine against cancer in many applications. In this review, we will discuss the state of the art of currently available solutions to generate and culture tumor spheroids and organoids, their advantages and limitations, as well as their applications.

## 2. Current Source to Generate and Culture In Vitro Tumor Spheroids and Organoids

Spheroids and organoids can be generated from different cell and tissue sources, including primary cells/biopsies [12,13,14], cancer stem cell lines, and tumor cell lines [15,16] (Figure 1). As each of these sources has its own advantages and limitations, researchers must carefully consider which source and method is the most appropriate for a specific application. The general advantages and limitations of each cell source are summarized in Table 1.

### 2.1. Spheroids from Tumor Cell Lines

Cancer cell lines are isolated from a patient’s tumor and selected for their growth in culture. These cell lines can be propagated indefinitely in culture, which allows for the generation of large numbers of spheroids for experimental purposes. One of the main advantages of using cancer cell lines to generate spheroids [17,18] is that they are readily available and can be purchased from various sources, such as the American Type Culture Collection (ATCC). This makes them easily accessible to researchers. The generated spheroids can be used to study the effects of drugs acting directly on cancer cells [19,20]. Very recently, Husain Yar Khan et al. tested the combination of a new anti-cancer drug (CPI-613) with radiation using a spheroid model of pancreatic cancer (MiaPaCa-2 and Panc-1 cell lines) [21]. Their results show that a combination of radiation with CPI-613 significantly inhibits pancreatic cancer cell growth compared with radiation alone. Moreover, in the context of drug testing in cancer, Sang-Eun Yeon et al. previously showed that Panc-1 spheroids may represent an effective 3D model for anti-pancreatic cancer drug screening [22]. In the context of lung cancer, A. Zuchowska et al. presented a protocol for the generation of spheroids from malignant (A549 cell line) and non-malignant cells with high viability, suitable for drug cytotoxicity studies [23]. In 2018, E. O. Mosaad et al. published a study using a new 3D prostate cancer spheroid platform to perform high-throughput drug screening [24]. Their results demonstrated that spheroids, in comparison to 2D monolayers, are not hypersensitive to chemotherapy, providing a superior model for the evaluation of single and sequential drug treatment. Bladder cancer spheroids were also generated from the RT4-cell line, as a model of drug screening and showed that the obtained spheroids are suitable for drug screening/cytotoxicity assays [25]. In addition to drug testing, spheroids from cell lines are also used in studies dedicated to the investigation of drug resistance [26]. Fan et al. demonstrated that spheroids formed from lung (A549) and pancreatic (PANC1) adenocarcinoma cell lines showed a higher resistance to anti-cancer selenite than cells in 2D [27]. Similarly, in the context of the study of anticancer drug resistance, a German team demonstrated the proof of concept of a high-throughput device allowing the culture, screening, and handling of pancreatic and colorectal spheroids [28]. Renal cell carcinoma spheroids were also used to describe and study the profile of stem cell-like cancer cells, which can also be responsible for metastatic spread and drug resistance [29]. Exploiting a high-throughput automated platform for spheroid culture using a polydimethylsiloxane (PDMS)-based hanging drop array (PDMS-HDA), a study showed that spheroids from several cancer cell lines such as breast, prostate, and colorectal cancer can be generated and show different levels of sensitivity to chemotherapeutic drugs and radiation as compared to 2D cultures [20]. However, there are common limitations in methods using cancer cell lines to generate spheroids [30]. Cancer cell lines are not representative of all patients, since each patient’s molecular signature is unique [31,32]. Moreover, the cancer cell composition is heterogeneous in vivo and it is expected that only some clones with a selective advantage will grow. Furthermore, cancer cell lines can acquire genetic and epigenetic changes over time, which may not reflect the original tumor cell characteristics, and therefore affect the results obtained from the spheroid model. Additionally, cancer cell lines do not represent the TME.

### 2.2. Organoids Derived from Primary Cells and Patients’ Biopsies

One alternative to cell line-derived spheroids is the generation of organoids from primary tissues such as patient-derived biopsies. Such 3D structures are known as Patient-Derived Organoids (PDOs). PDOs have been successfully generated from several cancer types such as breast cancer [33,34,35,36], lung cancer [37,38], gastro-intestinal cancer [39], gastroesophageal cancer [40], pancreatic cancer [41], ovarian cancer [42], prostate cancer [43], glioblastoma [44], liver cancer [45], colorectal cancer [46], retinoblastoma [47], and also bladder cancer [48]. The patient’s tumor sample may be a solid surgical resection material [33], punch or fine-needle aspiration biopsy [39,41,49,50,51], or biological fluid biopsy [42,51,52,53]. The current process of generating organoids from primary biopsies typically involves the resection of a small part of the tumor, which is cut into smaller pieces, and then mechanically and/or enzymatically dissociated for culture under conditions that promote self-organization into a 3D structure. The culture conditions such as the choice of media and method of organoid generation depend on the type of tumor. These conditions often include the use of specific growth factors and extracellular matrix components that are important for maintaining the proper cellular organization and function. The medium commonly used in many studies is DMEM/F12 medium, due to its richness in nutritional factors and because it is suitable for clonal culture. The latter or another medium is supplemented with small molecules, amino acids, cytokines, growth factors, and other supplements depending on the application and the cancer type [54]. One of the main advantages of using primary biopsies is a more accurate representation of the patient’s specificity, including the molecular signature [47]. PDOs have gained popularity as an effective and rapid 3D tool that better reproduces many tumor features, including the specific genetic and molecular diversity of the original host, hypoxia, nutrient diffusion, and metabolism [55]. PDOs that can retain host-derived TME are a valuable candidate for holistic approaches to 3D immune-oncology TME modeling. However, these models are prohibited by immune cells rapidly losing viability before studies. In a recent study, Yawei Hu et al. described a new and rapid protocol for PDO generation and prediction of drug response [37]. However, obtaining primary cells can be challenging and often require a surgical procedure. Many primary cells of the biopsy have a limited lifespan in culture and are not able to be treated by enzymes and die rapidly, generating a 3D structure in which some cells are missing, notably cells from the TME. For example, in lung cancer, the generation of PDO has, on average, a success rate of only 40%. The failure of PDO generation was associated with the quality of the sample biopsy and also the collection technique [37]. Additionally, variations in growth depend on the source of the biopsy, which can affect the outcome of the studies performed. Another limitation of PDO is the spatial heterogeneity of the tumor of origin; the biopsy may not fully capture the complexity and diversity of the tumor because the sample patient biopsy is often limited [56,57,58,59,60]. Then, the obtained PDO can represent the molecular signature and/or the TME of only a fraction of the original tumor. Sometimes, researchers may have to generate multiple PDOs from different regions of the patient’s tumor in order to study the effects of drugs and other therapies on the entire tumor, which is not easily feasible. Tumor composition and molecular signature vary not only within a patient, but also between patients. If this heterogeneity is exploited for personalized medicine, researchers may need to generate PDOs from multiple patients if their goal is to study the global effects of drugs and other therapies on different types of cancer. In practice, it can be technically, ethically, and financially difficult for researchers to obtain multiple samples from several patients. Therefore, exploiting tissue or cell banks to generate PDO can help to address the problem of patient-to-patient variability [57,58]. Biobanks exist for PDO generation [61] with material from breast cancer biopsies [34,35], pancreatic cancer [62], and colorectal cancer biopsies [63,64,65]. In the context of breast cancer, Norman Sachs et al. generated, in 2018, more than 100 primary and metastatic organoids [35,66]. More recently in 2022, Dan Shu et al. generated a new organoid bank from 17 patients [34]. Biobanks can be used to validate the results obtained from organoids generated from a smaller number of patients. However, biobanks can also have their own limitations [67]. Samples are usually collected at a specific time point, and may not reflect the temporal changes that occur in a patient. Additionally, the samples may not be stored or processed in a standardized manner, which can affect the quality and composition of the samples and make it difficult to compare data.

### 2.3. Spheroids and Organoids from Genetically Modified Cells

Cancer originates from different and variable genetic and epigenetic aberrations. This variability is recorded in the Cancer Genome Atlas, which describes more than 15,000 tumors [68]. Recently, an important effort has been made to identify the different genetic and epigenetic networks responsible for the cancer development [69]. Genetically modified cells can be used to generate organoids by culturing them in a 3D environment that promotes self-organization. Genetically modified cells to generate cancer organoids include the ability to study specific genetic pathways and the effects of specific mutations that are found in a patient. This can provide a deeper understanding of how genetic changes contribute to the development and progression of cancer and can be used to develop new targeted therapies. Meanwhile, cluster regularly interspaced short palindromic repeats-associated protein 9 (CRISPR/Cas9) technology has revolutionized genome editing and is applicable to the organoid field. CRISPR/Cas9-based gene modification allows the engineering of organoid models of cancer through the introduction of any combination of cancer gene alterations to normal organoids, including knock in (KI) or knockout (KO) of oncogenes or tumor suppressor genes (TS), and gene repression or activation. Multiplex editing by lentivirus or plasmids has been successfully tried [70,71,72,73,74,75,76]. Remarkable results have been achieved in this field. Artegiani et al. introduced BAP1 loss-of-function by CRISPR/Cas9 in normal human cholangiocyte organoids. They observed that BPA1 mutant organoids lost their organization and polarity and cells became more motile and fused with other organoids [77]. These features recapitulated the hallmarks of cancers. Interestingly, after restoring the catalytic activity of BAP1 in the nucleus, they observed a reversion of organoid morphology and molecular alterations to a level similar to WT organoids. CRISPR/Cas9 genome editing was also used in two independent studies in order to model multistep tumorigenesis in normal human intestinal organoids. Matano et al. generated intestinal organoids from normal human intestinal epithelium harboring mutations in the tumors suppressors genes APC, SMAD4, and TP53 and in the oncogenes KRAS and PIK3CA (five hit APC, KRAS, SMAD4, TP53, and PIK3CA (AKSTP)). They demonstrated that organoids carrying these mutations grow independently of all niche factor supplementations and formed tumors after implantation into immunocompromised NOG mice [78]. The same approach was used by Drost et al. to target APC, KRAS, TP53, and SMAD4 genes that designed AKPS in human small intestine and colon organoids. In their study, the authors demonstrated that the quadruple mutant organoids grew and were able to form invasive carcinomas upon subcutaneous xenotransplantation [79]. The approach of genetically engineered-based tumorigenesis has been also extended to normal primary human gastric organoids coupled with gene–drug interaction screens. CRISPR/Cas9-mediated ARID1A/TP53 dual KO organoids mirror several clinical–pathologic features of ARID1A-mutant gastric cancer. A high throughput drug screening revealed that ARID1A deficient gastric organoids were uniquely sensitive to a small molecule inhibitor of BIRC5/surviving [80]. A novel model of CRSIPR-Cas9-engineered TP53-CDKN2A dual KO human normal gastroesophageal junction (GEJ) organoids was generated for the first time. TP53-CDKN2A KO in GEJ induced morphological dysplasia as well as pro-neoplastic features in vitro and in vivo [81]. Takeda Haruna et al. used genetically defined benign tumor-derived organoids carrying two frequent gene mutations (APC and KRAS mutations; AK organoids), which mainly contribute to the disease progress in the early stage of colorectal cancer (CRC) [82]. They demonstrated that AK organoids recapitulate human CRC when transplanted into mice. In an alternative study published in 2016, Verissiomo CS and colleagues tested EGFR and MEK inhibitors on a large panel of CRC organoid lines in order to determine the effect of RAS-mutation status on the sensitivity to these drugs [83]. They demonstrated that the introduction of a KRAS G12D mutation by CRISPR/Cas9 resulted in a loss of drug sensitivity compared to WT. Recently, Neel’s group generated multiple high-grade serous tubo-ovarian (HGSC) cancers by engineering mouse fallopian tube epithelial organoids using lentiviral gene transduction and CRISPR/Cas9 mutagenesis [84]. HGSC models exhibit mutational combinations seen in patients and present several expected but other unanticipated sensitivities to small molecule drugs. In another report, ovarian cancer 3D spheroids were subject to genome editing using CRISPR/Cas9 to inactivate TIMP-2. These modified 3D spheroids exhibited low MMP-2 expression and high MMP-14, TWIST1, and SNAIL expression, enhanced proliferation, migration, and invasion. TIMP-2 KO spheroids were resistant to paclitaxel and formed long sheath-like cell aggregates, which showed enhanced proliferation and expression of the invasion marker KRT14 [85]. Lastly, a study suggested that customized therapy targeting ALDH1 could reduce resistance to chemotherapy and improve the survival rate of ovarian cancer. Consistent with this, ALDH1 inhibition by CRISPR/Cas9 effectively blocked the proliferation and survival of OC spheroids [86]. To summarize, CRISPR/Cas9-generated organoid models can recapitulate the molecular and pathohistological characteristics of human diseases and especially the multistep tumorigenesis from normal cells to malignant cells. Although CRSPR/Cas9 has simplified genetic engineering, there is a considerable margin of error during this process. In order to improve this, a new genetic tool for targeting specific genes in human organoids called CRISPR-Cas9-mediated homology-independent organoid transgenesis (CRISPR-HOT) was pioneered by Artegiani et colleagues. This technique was applied to fluorescently tag and consists of visualizing subcellular structural molecules for rare intestinal subset cells by generating reporters. This technique can be applied to study cell fate, differentiation, and disease development and can be used to visualize any type of gene or cell. This CRISPR-HOT was tried in liver organoids for knock in of cadherin and beta-tubulin genes to label the hepatocyte membrane and mitotic spindle, respectively, for the monitoring of hepatocyte division [87]. Finally, there are also limitations to using genetically modified cells to generate cancer organoids. The genetic modification process itself can introduce further complexity and variability to the organoids, making it harder to draw conclusions from the results obtained.

### 2.4. Cancer Organoids from Pluripotent Stem Cells

Cancer organoids can be generated from human Pluripotent Stem Cells (hPSCs), which can be embryonic stem cells or induced pluripotent stem cells (iSCs) [88]. The main advantage of using hPSCs is that they can be genetically modified to model specific organs and diseases, including cancer. As previously described, the CRISPR/Cas9 technology can also be applied to introduce specific mutations that are found in human tumors. These modified cells can then be differentiated into organoids that mimic the cellular and molecular features of the corresponding tumor type. Genetically modified organoids from pluripotent stem cells are well suited to investigate cancer development and progression, as well as to be a response to treatment. Brain tumors are among the deadliest and most aggressive cancers worldwide. In a study published in 2018, Shan Bian et al. used genetically modified hPSCs to generate brain cancer organoids [89]. The researchers introduced the mutation into the cells after the first neural induction step by transposon-and CRISPR/Cas9-mediated mutagenesis. These organoids allowed the exploration of the underlying mechanisms of tumor progression and the evaluation of the efficacy of therapies directed at specific genetic mutations. The study also demonstrated that these models are superior to traditional brain tumor spheres and 2D glioblastoma cell cultures as they allow interactions between tumoral and non-tumoral cells within the same system to be revealed. Similarly, in the field of glioblastoma, a study led by Junko Ogawa and colleagues established a cancer model of glioma in human brain organoids for investigating cancer progression, specifically the invasion phase [90]. To generate this model, the authors applied CRISPR/Cas9 technology to integrate an HRasG12V-IRES-tdTomato construct into the TP53 gene by homologous recombination in the H9 hPSC line. Interestingly, the mutant cells quickly became invasive and destroyed the surrounding organoid structures. The invasive nature of these tumor cells was further supported by transplanting the cell into immune-deficient animals. The cells generated by the organoids displayed gene expression profiles that were consistent with human glioblastoma, further illustrating the potential of using organoids as a platform to replicate key features of malignancy. More recently, Markus Breunig et al. developed a pancreatic duct-like organoid (PDLO) model from human pluripotent stem cells to study pancreatic cancer formation from a genetically defined background [91]. They developed a model of pancreatic carcinogenesis by inducing the expression of oncogenes GNAS and KRAS using piggyBac transposon-based vectors combined, or not, with CDKN2A KO by CRISPR/Cas9. Indeed, after PDLO transplantation in mice, they showed that PDLO expressing GNAS formed large oncogenic cystic structures, whereas KRAS alone, induced a diverse range of abnormal growths. However, when KRAS was combined with the loss of CDKN2A, it led to the development of malignant and dedifferentiated pancreatic ductal adenocarcinomas. These results highlighted the possibility to use this PDLO to model the genetic changes that lead to pancreatic cancer formation. Another study in the same field showed that GNAS oncogene expression induces cystic growth more efficiently in ductal organoids than in acinar organoids, while KRAS was more effective in modeling cancer in vivo when expressed in acinar organoids compared to ductal organoids derived from pluripotent stem cells [92]. Using also the activation of KRAS, Antonella F.M. Dost et al. developed an organoid system from human iPSC-derived lung epithelial cells to model early-stage lung adenocarcinoma, illustrating another time that the potential of cancer organoids is derived from hPSCs [93]. The study demonstrated that alveolar epithelial progenitor cells that expressed oncogenic KRAS showed a decrease in the expression of maturation genes. This research provides a comprehensive dataset that differentiates normal epithelial progenitor cells from those in early-stage lung cancer, making it easier to identify targets for KRAS-driven tumors. While organoids generated from PSCs have a great potential to model diseases, to identify new therapeutic targets and to test the efficacy of different drugs on the tumors, there are some technical and ethical challenges that need to be considered. For example, there is a risk of contamination with undifferentiated stem cells, and the organoids may not fully replicate the complex cellular interactions of a living organism; therefore, they may not fully mimic the disease in vivo. Moreover, there are some ethical concerns associated with the use of embryonic stem cells. Finally, the capacity to genetically modify hPSCs has enormous potential for cancer modelization through cancer organoids. However, while it has been relatively easy to edit immortalized human tumor cell lines [94], it is not as easy with hPSCs [95,96,97].

### 2.5. Organoids Made from Several Cell Sources

Organoids can be generated from a mixture of different cell sources, including primary cells, endothelial cells, stromal cells, or tumor cell lines. Combining different cell types in a mixture can create an environment that closely mimics the complexity of the in vivo TME. Yeonhwa Song and colleagues developed an organoid model using several cell types such as hepatocellular carcinoma cells, fibroblasts (WI38 cell line), hepatic stellate cells (LX2 line), and the endothelial primary cell line HUVEC to establish a liver fibrosis model [98]. The aim of this study was to identify potential mechanisms and inhibitors of liver fibrosis, suggesting that anti-fibrosis drugs may improve tissue permeability to support the delivery and efficacy of anti-cancer drugs. This model has the potential to offer an efficient strategy to identify new drugs and targets in an accurate organoid model close to the in vivo situation. Similarly, to recapitulate bone marrow in both normal and tumoral conditions, many different studies have been performed [99,100]. Recent studies have developed bone marrow organoids (BMOs) [101,102,103]. Recently, we developed a BMO model that mimics several structural and cellular features of native bone marrow (BM) [103]. These BMOs were formed from Mesenchymal Stromal Cells (MSCs) and endothelial cells, modeling the migration and integration of leukemic cells within the BM, highlighting the potential of this model as an easily accessible and scalable preclinical model for evaluating the efficacy of potential drugs in personalized medicine. Another BMO model has been developed recently, using hPSCs differentiated into mesenchymal, endothelial, and hematopoietic lineages. It recapitulated the stroma, sinusoids that form lumens, and myeloid cells, including proplatelet-forming megakaryocytes [102]. This model can also sustain primary cells from patients of myeloid and lymphoid blood cancers within the context of their microenvironment and represents a crucial ex vivo tool for evaluating new therapeutics. One advantage of using a mixture of cell sources to generate organoids is that it allows researchers to study the interactions between different cell types in the TME and their impact. For example, H. Zhao et al. investigated the co-culture of tumor-infiltrating fibroblasts with oral squamous cell carcinoma organoids. They showed that the co-culture with fibroblast promoted the stemness properties of the primary carcinoma [104]. However, generating organoids from a mixture of cell sources can be technically challenging [105], especially if the cell types that need to be mixed have different culture requirements or different growth rates. Additionally, the organoids may not fully replicate the complex cellular interactions of a living organism, and there can be variations depending on the cell source used and the method of preparation.

### 2.6. Organoids Including the TME

Cancer organoids have a limitation in their application when studying the impact of the surrounding TME, as they often lack the representation of immune cells [106,107]. Recent studies have therefore focused on the development of cancer organoids that include immune cells. These models have been used to study the interactions between cancer cells and the immune system, and to test immunotherapies. By including immune cells and components, these organoids better mimic the patient’s TME and can provide valuable insights into cancer biology including the mechanisms of immune evasion and resistance [108]. They can also be used to identify new targets for immunotherapy and to evaluate the efficacy of new drugs. The types of immune cells that should be included in cancer organoids depend on the research question and the type of cancer being studied. Some common immune cells included in cancer organoids include T cells (cytotoxic or regulatory), NK cells, macrophages, dendritic cells (DCs), myeloid-derived suppressor cells (MDSCs), or neutrophils. The immune cells must be derived from the same patient to study the patient-specific immune response and prevent allogeneic responses. A study presented a method successfully preserving several endogenous immune cell types including T cells, macrophages, B, and NK cells, in PDOs [64]. These organoids were derived from more than 100 human tumor biopsies of skin, kidney, and lung cancers and were used to study the effects of drugs that inhibit immune checkpoint proteins such as PD-1 and PD-L1 [64]. Another study performed a co-culture of two different human lung cancer cell lines (A549 and Calu-6) alone or together with fibroblasts and peripheral blood mononuclear cells (PBMCs) [109]. The authors showed that when the cancer cells were cultivated with fibroblasts, the infiltration capacity of cytotoxic T lymphocytes was increased, demonstrating that (i) immune cells could be added to the cancer 3D model, and (ii) the TME significantly impacted immune cell infiltration and activation. In addition, the inclusion of fibroblasts in cancer organoids has been shown to lead to a change in the type of T-cells present, with a greater proportion of activated ones [110]. There are still a number of limitations to the general use of organoids that include immune cells [61]. One of the main limitations is the complexity of the TME and the difficulty to define or keep its original composition in culture. Moreover, the use of organoids to study cancer immunotherapies is still relatively new, and there is still much to be learned about how these models can be used to predict responses to therapy in patients. Then, there is always the concern that the organoids may not fully represent the in vivo biology of the tumor and its immune system. Thus, methods that will ensure the complete maintenance of the original TME will be the better options.

## 3. Methods for the Generation and Culture of Cancer Spheroids and Organoids

### 3.1. ECM (Extracellular Matrix)-Free Methods for Cancer Spheroids and Organoids

ECM-free methods refer to technologies used to derive and cultivate cancer organoids without any external matrix. These methods rely on the ability of cells to self-organize and produce their own extracellular matrix and cell-to-cell and cell-to-matrix interactions. Some examples of scaffold-free methods include liquid overlay spheroid formation, hanging drop culture, and micro-patterned surfaces (Figure 2). Scaffold-free methods for cultivating cancer organoids have several advantages. They promote the natural self-organization of cells into 3D structures, which is similar to the native tissue architecture. This helps to mimic closely the interactions between cells and the extracellular matrix. These methods are generally more flexible in terms of types of cancer cells that can be cultured, since they do not rely on the use of specific extracellular matrix proteins. This can make it easier to culture a wide variety of different cancer cell types. Another advantage of scaffold-free methods is that they are more cost-effective and easier to scale up for large-scale experiments or for use in clinical applications. Finally, Scaffold-free methods also allow a more controlled environment, such as the oxygen and nutrient levels, which are important for the survival and growth of the cells. The choice of the method depends on the type of organoid and the research question. Each system has its own advantages and disadvantages, and researchers must carefully consider which system is best suited for their specific application. Some of the most common methods include the hanging drop method, the use of low attachment/repellent plates (liquid overlay spheroid formation), forced aggregation into microwells, magnetic levitation, rotary cell culture systems (bioreactor/spinner), and microfluidic device systems (Table 2).

#### 3.1.1. Hanging Drop Method

This method was originally described by Ramsey Foty [111]. It involves a small droplet of a cell suspension suspended in air, typically using a small piece of filter paper or a coverslip (Figure 2A). The droplet is then placed in a humidified chamber and incubated at the appropriate temperature and conditions. The cells in the droplet will aggregate and form a spherical structure. One of the main advantages of the hanging drop method is that it allows for the formation of spheroids in a minimal volume of medium, which can help to reduce the number of reagents needed and can also help to reduce the amount of waste generated. However, after the formation of spheroids, they must be transferred to a cell culture plate for further cultivation in a submerged medium. Entaz Bahar and her colleagues used the hanging drop method to generate ovarian cancer spheroids of 1500 to 2000 cells to study the effectiveness of a combination therapy on resistant ovarian cancer cell lines [112]. They targeted Twist, a known oncogene that contributes to cisplatin resistance in ovarian cancer cells, by using a knockdown approach. The study demonstrated that blocking the DNA-damage response in cisplatin-resistant OC cells through Twist knockdown sensitized these cells to concurrent treatment with cisplatin and niraparib, a PARP inhibitor, in both 2D and 3D cell culture. This study highlighted the potential of using scaffold-free 3D models to study resistance mechanisms and test new therapeutic combinations in ovarian cancer. In colorectal cancer, this method was also used to study resistance mechanisms in cancer stem cell subpopulations [113]. This study used the hanging drop method to create spheroids from two different cell lines, HT-29 and Caco-2, and found that both cell lines formed spheroids that were enriched with cancer stem cells. This was indicated by the upregulation of genes associated with stemness. The study demonstrated that spheroid culture is a reliable and cost-effective way to mimic the complexity of in vivo tumors, including self-renewal, drug resistance, and invasion, for in vitro research on colorectal cancer stem cells. Another study previously mentioned [110] used the hanging drop method to co-culture human lung cancer cell lines A549 and Calu-6 with the human fibroblast cell line SV80 for 10 days to form microtissues. After 10 days, peripheral blood mononuclear cells were added to study the TME. The method was also successfully used in a context investigating the potential benefits of CIGB-300, an antitumor peptide with a novel mechanism of action targeting CK2-dependent signaling pathways, in treating non-small cell lung cancer [26]. The hanging drop method has important limitations and drawbacks: it is time-consuming and involves a small droplet of cell suspension in the air, which strongly limits the number of cells that can be cultured in each droplet; and it is not easily scalable. Moreover, the size and shape of the spheroids formed using the hanging drop method can be affected by factors such as the size of the droplet and the initial seeding density of the cells. Moreover, the hanging drop method is not suitable for cell types requiring time to form spheroids [25,110,114]. For example, a study comparing the hanging drop method to the forced floating method (ultra-low attachment plate) using an RT4 human bladder cancer cell line as a model, demonstrated that the hanging drop method was less suitable to generate RT4 spheroids for drug screening/cytotoxicity assays [25].

#### 3.1.2. Non-Adherent Plates and Forced Floating Method

The method is based on the concept of using a surface that repels cells, rather than promoting attachment, thus forcing cells to clusterize (Figure 2B). There are several types of non-adherent plates that can be used to culture cancer organoids, including Ultra-Low Attachment (ULA) plates and hydrogel-coated plates [25]. This method is also known as liquid overlay spheroid formation or repellent plates. ULA plates are made from a variety of materials, such as plastics or glass, and have a surface that is treated to prevent cells from attaching and spreading. The ULA method is simple and easy, as it only requires a special type of plate. This method was used for several cancer types, such as pancreatic cancer [28], glioblastoma [115], colorectal cancer [113], bladder cancer [25], and human cervical cancer [114]. To model pancreatic cancer, MiaPaCa-2 cells were plated in ULA plates [21] to investigate whether CPI-613, a small molecule inhibitor of mitochondrial metabolism, could be used to sensitize pancreatic ductal adenocarcinoma to radiation therapy. The study found that a combination of radiation with CPI-613 significantly inhibited tumor cell growth compared to radiation alone. Similarly, brain tumor spheroids were formed by seeding around U87-MG cells in a ULA plate [115] and embedded into Matrigel to monitor the invasion of the extracellular matrix. Other groups used ULA plates to generate colorectal cancer [113] or pancreatic cancer spheroids [28] and cervical carcinoma spheroids [114]. Finally, the ULA plates were compared with the hanging drop method by using the RT4 human bladder cancer cells. Both methods were able to generate spheroids with similar characteristics, but the ULA method was found to be more suitable and straightforward for generating spheroids for drug screening and cytotoxicity assays. The study also found that 3D cultures exhibited higher resistance to the drug doxorubicin than 2D cultures [25]. However, there are still limitations and challenges with the ULA method such as inadequate oxygen and nutrient supply to the center of the spheroids, leading to necrotic centers due to the cell culture in a submerged medium [116]. Additionally, some cell types may still adhere to the support and form monolayers, despite the surface being treated to reduce cell attachment. The method also presents a lack of standardization, leading to spheroids with a very high heterogeneity in size and shape. To overcome this limitation, other methods based on forced aggregation into microwells were developed [117].

#### 3.1.3. Forced Aggregation into Microwells

The method involves the use of microwells, which are small, cylindrical wells typically formed in a polymeric substrate. Cells are suspended in a liquid medium and added to the microwells. The cells will sediment into the bottom of the microwells and then aggregate to form a 3D structure (Figure 2C). The microwell architecture allows for the formation of multiple organoids in a single well. This method allows for the formation of organoids on a relatively large scale, and also allows for a more consistent and reproducible size of organoids. One of the main advantages of the microwell method is that it allows for the formation of multiple organoids in a single well, which can be useful for high-throughput screening. Additionally, the microwells allow the control of the size, shape, and composition of the spheroids and organoids [118]. The number and composition of the cells that form spheroids in each microwell depend only on the properties of the cell suspension, making it possible to generate spheroids of arbitrary size and composition. The high number of microwells also allows for the high production levels of spheroids through a straightforward, non-labor-intensive process, making it a useful tool for researchers in the field of tumor spheroid biology. This technique was used to generate tumor spheroids in a well-controlled and reproducible manner with various cancer types such as colorectal (HT-29 cell-line), prostate (LNCaP cell-line), or esophageal cancer (TE6 cell-line). However, this system has some limitations such as overgrowth, which can be prevented by starting with smaller spheroids or using larger microwells. Moreover, the spheroids and organoids produced cannot be maintained in their respective microwells due to their attachment, and must be transferred rapidly to a new ULA plate. For this reason, a method was developed with microwells made of PDMS [22]. Pancreatic cancer spheroids were successfully cultured by this system and are useful for drug development and identifying resistance mechanisms. In a similar way, a Japanese team used a micro molding technique to create PDMS-based microwells with different widths, and used them to generate multicellular spheroids of varying sizes using human hepatoblastoma HepG2 cells [119]. The main limitation of forced aggregation is the stability of spheroids in their respective microwells. Indeed, due to small vibrations in the incubator, the spheroids can very easily rise to the surface of the microwells and exit rapidly to fuse together with other ones. To overcome this limitation, new microwells with a 36 μm mesh at their top were developed, large enough to allow passage of single cells, but small enough to retain spheroids into microwells. Another recent study described an original method for forming layered tumor spheroids using ultrasound. The technique involved microwells and a circular transducer to introduce ultrasonic standing waves into the microwells. The cells are seeded above the microwells and allowed to sediment to the bottom. The radial forces trap the cells in an aggregate in the center of the microwell to form the spheroid [120]. The method was used to generate and cultivate ovarian cancer spheroids using the OVCAR-8 cell line, resulting in spheroids that mimic the structure and cellular distribution of solid tumors. Microwell-based methods are limited by the fact that cells are clustered and cultured in a submerged medium, which can limit oxygen diffusion and lead to hypoxia and cell death, making it difficult to maintain high cell viability [117,121].

#### 3.1.4. Magnetic Levitation

Magnetic levitation has been used to generate cancer spheroids or organoids by suspending cells in a magnetic field. In this technique, cells are suspended in a solution containing magnetic beads, and then exposed to a magnetic field, which causes the beads and cells to levitate (Figure 2D). The cells then form spheroids or organoids in the absence of any external mechanical force. This method has been used to study various types of cancer, including breast [122,123], lung, and pancreatic cancer [27]. One of the key advantages of using magnetic levitation for generating cancer spheroids or organoids is that it allows for the formation of homogenous and uniform cell aggregates. Additionally, magnetic levitation can be used to create dynamic and versatile microenvironments for cells, notably by applying rotating magnetic fields that can mimic the mechanical forces of blood flow, or by using magnetic fields to manipulate the position of cells in real time. For example, a group used the Bio-Assembler™ System from n3D Biosciences, Inc (Houston, TX, USA), to construct a breast tumor spheroid model [122]. Finally, these two studies have shown promising results of magnetic levitation for cancer spheroids, which is considered to be a valuable tool in the field of cancer research. However, it is a relatively new technology and more research is needed to fully understand its potential uses in cancer research. The cost of the equipment and the magnetic beads can be high. It may be difficult to scale up to larger numbers of cells or larger cell cultures. Depending on the type of magnetic beads used, it may not be compatible with all types of cells, or it may require additional steps to remove the beads after the cells have formed spheroids or organoids [117].

#### 3.1.5. Rotary Cell Culture System (Bioreactor/Spinner)

In the context of cancer research, bioreactors were used to culture organoids in a controlled environment for various cancer types, such as neuroblastoma [124], breast cancer [125], non-small lung cancer [126], and hepatocellular carcinoma [127] (Figure 2F). A study looked at the aggregation kinetics of neuroblastoma cells and the formation of organoids, and specifically examined the effect of oncogene MYCN amplification on cell behavior [124]. The results showed that the MYCN-amplified cell line aggregated more quickly and formed a different morphological structure compared to the unamplified cell lines. This system was found to be a rapid and reproducible assessment of in vitro behavior, and the parameters measured correlated with the malignant potential in terms of MYCN amplification. Another study used an equivalent system to produce breast cancer spheroids by co-culturing breast cancer cell lines and fibroblasts [125]. The cells interacted spontaneously with each other, resulting in the formation of spheroids, which consistently showed the expression of various proteins and cellular structures similar to those found in the breast cancer tissue. More recently another bioreactor system was presented to generate non-small lung cancer spheroids [126]. Finally, in 2021, hepatocellular carcinoma spheroids were cultured in bioreactors after being generated in microwell plates, as described in Section 2.3. In this study, they developed an accurate method for in vitro genotoxicity testing [127]. The researchers used 21-day-old spheroids made from human hepatocellular carcinoma cells grown under controlled conditions in a dynamic bioreactor. The spheroids were exposed to non-toxic concentrations of two pollutant compounds. The results showed that these pollutants significantly increased DNA strand breaks, as well as gene expression changes related to metabolism and DNA damage. The study concluded that the spheroid model in the dynamic bioreactor was a sensitive and promising method for genotoxicity testing and could provide reliable results for environmental exposure assessments. 

The main advantages of using bioreactors [128] for cancer organoid cultures include the ability to generate organoids in a controlled environment, the ability to mimic the in vivo microenvironment of tumors, and the ability to culture very large numbers of organoids. Indeed, bioreactors can provide a range of physical and chemical cues, such as oxygen tension, pH, temperature, and nutrient concentration, which can be controlled to mimic the in vivo microenvironment. However, generating uniform spheroids with a specific size and shape using bioreactor systems can be challenging. Moreover, these systems can be relatively expensive and require specialized equipment and training to operate [117].

### 3.2. ECM-Based Methods for Cancer Spheroids and Organoids

In concordance with the fact the ECM is a major component of the tumor microenvironment (TME) with tumor-inducing capabilities, several ECM -3D models have been engineered to study the interactions of the TME components. Tumor cells cultured in decellularized native tissues or natural/synthetic biomaterial scaffolds that provide proper cell adhesion, differentiation, and migration properties, and closely mimicked the cell–ECM interactions [129,130,131]. Therefore, it is crucial to integrate an appropriate ECM into tumor organoid cultures. To generate spheroids and organoids incorporating ECM, researchers typically mix cells with a cold ECM as previously described [132]. The mixture is then deposited in small droplets, or domes, typically within a well of a culture plate. The use of a small volume of ECM droplets helps to create a controlled environment and prevent the formation of large, unmanageable structures. Once the droplets have been deposited, the plate is placed in an incubator at 37 degrees Celsius, allowing the ECM droplets to solidify and harden. This process typically takes a few minutes to several hours, depending on the specific ECM and the desired size and complexity of the spheroids or organoids. As the ECM solidifies, it creates a three-dimensional structure that supports the growth and differentiation of the cells within it. Over the last decade, the most commonly used matrix for tumor organoid cultures has been basement membrane extracts (BMEs) [133], which will be the main focus of this review. BMEs have been commercialized under the name “Matrigel^®^”. It consists of basement membrane extract from Engelbreth–Holm–Swarm (EHS) murine tumors. BMEs include laminin, collagen IV, entactin, and heparan sulfate proteoglycans [134], as well as several growth factors such as fibroblast growth factor, epidermal growth factor, insulin-like growth factor 1, transforming growth factor beta, platelet-derived growth factor, and nerve growth factor. As a recent example, Badea et al. formed multiple MDA-MB-231 breast cancer adenocarcinoma multicellular tumor spheroid (MCTS) in a single well by including Matrigel^®^ in the culture media and observed a more uniform morphology and greater circularity compared with MCTS formed without Matrigel^®^ [135]. When Matrigel^®^ was compared to lower-cost and easier-to-handle extracellular matrix-derived products such as gelatin-alginate and collagen-alginate to model breast cancer, only the Matrigel^®^ ink (2% v/v) successfully induced MCF10A, MCF10A-NeuN, MDA-MB-231, and MCF7 epithelial breast cancer MCTS formation [136]. Although Matrigel and BMEs have provided a tumor-relevant environment for human tumor organoid cultures, it has limitations, such as poor control of mechanical properties, batch-to-batch variation, and the risk of animal pathogen transmission [137,138]. As a result, alternative scaffold systems, including synthetic and natural hydrogels, have been developed to address their limitations [139]. Collagen-based matrices, which mimic the microenvironment of cancer cells and regulate invasive cellular behaviors, have been widely used in 3D cancer models [140]. Decellularized ECMs (dECMs) derived from various tumors, organs, tissues, and cell sheets, which preserve the organ- or tumor-specific biochemical profile of the original ECM, have also shown great potential in establishing tumor organoid models [138,139,140]. In vitro cancer models using patient-derived dECMs have demonstrated increased invasiveness and cellular dissemination compared to collagen-embedded models, and organ-specific metastasis can be recapitulated using dECMs from various organs. Native ECMs such as alginate, gelatin, hyaluronic acid (HA), and fibrinogen, as well as synthetic polymers such as polyethylene glycol (PE), polycaprolactone (PCL), and poly (lactic-co-glycolic) acid (PLGA), have also been used as matrices for tumor organoid modeling. Natural and synthetic polymers each have advantages and drawbacks, with natural polymers providing high biocompatibility and specificity, while synthetic polymers offer reproducibility and precise control over mechanical and chemical properties [140]. However, both natural and synthetic polymers have limitations in replicating the complexity of the native ECM, and further modifications are required to overcome these limitations [140]. Ongoing development and validation of composite hydrogels that accurately replicate dynamic tissue-specific cues will be crucial for practical applications of organoid cultures. This understanding of the ECM’s requirements for organoid growth can provide insights into organ development and disease progression, which could lead to new therapies and regenerative medicine applications [139,141].

## 4. Methodological Options in Front of Research on Anticancer Therapy

Different anti-cancer therapies, including chemo- and immunotherapies, have been successfully tested on PDO, making them a potentially universal strategy. Tumor organoid-based platforms provide an opportunity to achieve this goal by generating PDO and using them as an avatar of the patient to test personalized drug responses (Table 3).

### 4.1. Chemotherapy

A fully personalized approach for chemotherapy is currently lacking, and new predictive assays to help match patients to treatments are highly needed. Many results have shown that PDO might mirror clinical responses in individual patients. For example, a study derived PDO from metastatic lesions of colorectal cancer patients and showed that they could be used for chemotherapy testing such as resistance to irinotecan. Recently, 50 organoids from colorectal cancer liver metastases that capture intra and inter-patient heterogeneity were derived to evaluate whether PDO could effectively predict response to chemotherapy and clinical prognosis [142]. The 50 PDOs were exposed to 5-FU, CPT11, or oxaliplatin, frequently used first-line chemotherapeutic drugs, and a variable chemosensitivity to 5-FU, CPT11, or oxaliplatin monotherapy was shown, opening a potential application for personalized medicine. Another study tried to find associations between stem cell markers, patient survival, and resistance to therapy by using patient-derived colorectal cancer organoids [175]. The authors examined the expression of different stem cell markers in a cohort of PDO and correlated the expression with the sensitivity to 5-FU treatment. They identified Clusterin (CLU), a marker of the revival stem cell population, which was significantly enriched following 5-FU treatment, and its expression correlated with the level of drug resistance and a lower patient survival. PDOs from colorectal cancer with peritoneal metastases were also generated to test drug panels and identify specific drug sensitivities [176]. Genomic and drug profiling was completed within 8 weeks and a formal report ranking drug sensitivities was provided to the medical oncology team, resulting in a treatment change for two patients out of 28. Another study generated a pancreatic PDO library (N = 114 lines derived from 101 patients) and exposed a subset of these organoids to the five chemotherapeutic agents most commonly used [62]. PDO pharmacotyping revealed marked interpatient variability in the response to single chemotherapy and retrospective correlations with responses in patients. In another study, breast cancer PDOs were treated with Tamoxifen, and results demonstrated different sensitivities in agreement with clinical outcomes [35]. Finally, PDOs from ovarian tumors were generated to evaluate their capacity to predict clinical response [177]. Drug screening identified high responsiveness to at least one drug for 88% of patients in a period of 3 weeks. Hyperthermic intraperitoneal chemotherapy (HIPEC) is a method of administering anticancer agents directly while heating the abdominal cavity. The usefulness of HIPEC in combination with cytoreductive surgery for peritoneal pseudomyxoma and gastrointestinal cancer has been demonstrated [178,179]. Recently, PDOs have been used to study the efficacy of one HIPEC regimen compared with another [143,144].

### 4.2. CRISPR/Cas9

Recent studies have revealed that the CRISPR/Cas9 technique could remove oncogenic mutations, thus enabling colorectal cancer treatment. Genetically defined benign PDOs carrying two frequent gene mutations were shown to recapitulate the human disease in vivo after xenotransplantation in animals. Other researchers tested EGFR and MEK inhibitors on a large panel of colorectal cancer organoids in order to determine the effect of the Ras-mutation status on the sensitivity to these drugs. They demonstrated that the introduction of a KRASG12D mutation resulted in loss of drug sensitivity compared to WT. Recently, high-grade serous tubo-ovarian cancer organoids were obtained with mouse fallopian epithelial cells using a CRISPR/Cas9 mutagenesis. These tumorigenic organoids presented several expected, but also unanticipated, sensitivities to small molecule drugs, notably PARP inhibitors [84]. Ovarian cancer spheroids were also subjected to genome editing using CRISPR/Cas9 to inactivate TIMP-2, with the goal to understand its pathogenic role. These KO spheroids exhibited enhanced proliferation, migration, invasion, and resistance to paclitaxel [85]. Similarly, a study suggested that customized therapies targeting ALDH1 could reduce resistance to chemotherapy and improve the survival rate of ovarian cancer. Consistent with this finding, ALDH1 inhibition by CRISPR/Cas9 effectively blocked the proliferation and survival of ovarian cancer spheroids [86].

Interestingly, organoids can be isolated from an individual patient affected by a certain genetic monogenic disease, and the organoids can be genome-edited to correct the causing mutation. The first proof of concept of correction of a genetic defect by CRISPR/Cas9 genome editing in stem cells of human patients with a genetic disease was provided in 2013 [70]. Cas9-mediated homologous recombination with a wild-type version of the cystic fibrosis transmembrane conductor receptor (CFTR) locus was used to rescue the ΔF508 mutation in intestinal organoids isolated from two different CF patients. The authors performed the foreskin assay with transgenic lines. By live-cell microscopy, they observed rapid expansion of the organoid surface area in the corrected organoids, whereas swelling was absent in untransfected control organoids. Thus, this work provided a potential strategy for future gene therapy in patients.

Congenital dyskeratosis is a disease caused by mutations in the DKC1 gene, which results in impaired maintenance of telomere length. Individuals with this disease have a predisposition to developing leukemia and cancer, especially squamous cell carcinoma of the head and neck. hPSC-derived intestinal organoids from patients with this disease have been developed and used for Cas9-mediated recombination, and mutation correction resulted in a phenotypic rescue [145]. A group generated pancreatic ductal adenocarcinoma organoids [146] and found that Pin1 is overexpressed both in cancer cells and cancer-associated fibroblasts and correlates with poor survival. Targeting Pin1 function by CRISPR/Cas9 suppressed fibroblast proliferation, induced quiescence, and inhibited their ability to secrete a wide range of cytokines that promote cancer progression and prevent T cell recruitment into the TME.

KRAS is the most frequently mutated oncogene in human cancer. Programmable nucleases, particularly the CRISPR/Cas9 system, provide an attractive tool for genetically targeting KRAS mutations in cancer cells. S Sayed et al. demonstrated for the first time that oncogenic KRAS and TP53 base editing through CRISPR/Cas9 was possible in PDO from pancreatic ductal adenocarcinoma, colorectal cancer, and gastric cancer, and this correction impeded PDO growth [147]. Taken together, these studies showed that CRISPR/Cas9 has emerged as a versatile tool to activate/deactivate tumor suppressor genes and inactivate oncogenes, besides the correction of the disease-causing mutations. Further, the success rates of generating new organoid and cell-line models have increased substantially. These advances make it possible to apply new CRISPR/Cas9 technologies and small-molecule libraries to map the dependencies for each rare cancer type in the laboratory, leading to more robust therapeutic hypotheses. Although very promising, there are still only a few phenotypic rescues by the CRISPR/Cas9 editing system in human organoids. Clearly, transplantation techniques of organoids will have to be developed to allow the application of genome-edited, patient-derived human organoids in the clinic.

### 4.3. Adoptive Immune Cell Transfer Therapies

Adoptive cell transfer (ACT) has risen to be one of the fastest-growing immune-oncology fields in the past decades [180,181]. Next-generation immunocompetent cancer-on-chip models (iCoCs) based on in vitro microfabrication and microfluidics are able to precisely manipulate the special location of cells, of oxygen transport, of vascular barriers, and biophysical forces on physiological scales. iCoCs approaches are now used to study adoptive immune cell therapy against cancer. In a recent study, enriched tumor-reactive T cells showed their efficiency to kill in vitro tumor organoids [148]. In another study, Vδ2+ γδ T cells were consistently present in preparations of mammary ductal epithelial organoids, and they proliferated in response to zoledronic acid, an amino bisphosphonate drug [182]. These cells produced INF-γ and effectively eliminated in vitro mammary ductal epithelial organoids. A system called BEHAV3D was developed to study the interaction between immune cells and PDOs by means of imaging and transcriptomics analysis [149]. The BEHAV3D can track more than 150,000 engineered T cells co-cultured with PDOs. The authors also studied cancer metabolome-sensing engineered TEG cell (αβ T cells engineered to express a γδ TCR) behaviors. They detected a high variation in TEG-mediated killing efficacy in cultures derived from PDO biobanks. They also showed that the underlying behavioral and molecular mechanisms of cellular immunotherapy differed between different PDO cultures. These differences were also observed between individual organoids belonging to the same PDO cultures. This demonstrates that this platform captures inter- and intra-patient heterogeneity, a major obstacle for treating solid tumors. They demonstrated that type I INF can prime resistant organoids for TEG-mediated killing. CAR T cells are another type of genetically engineered T cell mostly used in ACT. Tumor organoid-on-a-chip models have been used to investigate the immunotherapeutic activities of CAR-T cells. A microdevice platform that recapitulated a 3D ovarian tumor allowed the exposition of tumor cells to CAR-T cell delivery and revealed cytotoxicity [150]. Other groups generated glioblastoma organoids to recapitulate inter- and intra-tumoral heterogeneity [57]. EGFRvIII-specific CAR-T cells were shown to penetrate organoids and a higher expansion of these cells was observed was accompanied by increased cytotoxicity.

Thus, this study demonstrated that organoids could be employed to test CAR-T cell immunotherapy in a clinically relevant timescale. More precisely, a study also showed that in a bladder PDO, co-cultured CAR T cells targeting MUC1 spontaneously migrated toward MUC1+ cells and induced a specific cell lysis [151]. A microfluidic breast cancer cell spheroid model was developed to study NK cell immunotherapies [183]. Using this system, the authors observed that NK cells alone or in combination with antibodies were able to directly penetrate the spheroids and destroy tumor cells in a matter of a few hours. NK cells can be genetically engineered to express CAR to increase their cytotoxicity function and to guide them toward tumors [184]. In glioblastoma, engineered NK-CAR cells targeting HER2, EGFR, and EGFRvIII were investigated in 3D tumors [152]. These 3D structures presented a dispersed morphology with an increase in the number of dead cells and in INF-γ, IL-6, and IL-8 secretion when co-cultured with the super-charged NK cells. A platform was also designed for the assessment of CAR-NK-92-mediated activity against patient-derived colorectal cancer organoids [185]. The authors proposed a sensitive in vitro platform to evaluate CAR efficacy and tumor specificity in a personalized manner. Indeed, they demonstrated that these CAR- engineered NK-92 cells were directed toward tumor organoids.

### 4.4. Immunotherapy

Dysregulations linked to the TME explain why conventional treatment often leads to relapse in most aggressive cancer types [186]. Therefore, predicting situations where patients will not respond to specific treatments is a keystone to improving cancer therapy. Over the past decade, immunotherapy has shown encouraging results in numerous types of cancer including melanoma and lung cancer. However, these treatments are not equally effective in all patient cohorts. This difference is due to the complex TME composition, which allows resistance to immunotherapy [187]. Therefore, modeling and analysis of the TME is an essential parameter to consider not only for the development of new therapies but also for the identification and stratification of patients who likely respond to them [188]. The emerging use of 3D cell culture models for testing immunotherapies could present an elegant alternative approach.

#### 4.4.1. Immune Checkpoint Inhibitors (ICIs)

Researchers have put a lot of effort into establishing relevant in vitro models to understand and determine the immunosuppressive TME and the numerous resistance mechanisms that are still unknown when ICIs are used. Several organoid models including glioblastoma, melanoma, colorectal carcinoma, and chondroma have been so far developed to study the response, efficacy, and specificity of ICIs therapies (Figure 3) [57,158,189].

The first strategy aimed to generate organoids that preserve endogenous immune cell types (NK cells, T and B cells, and macrophages) and stromal cells, thus mimicking the native TME. Neal et al. generated PDOs from more than 100 biopsies using an air–liquid interface (ALI) method [64]. For this method, tumor tissues are embedded in a type I collagen matrix on a transwell insert, ensuring adequate oxygen supply and the long-term preservation of the organoid structure. These PDOs preserved diverse immune cell types, such as T cells with the original tumor T cell receptor (TCR) repertoire and also stromal elements for weeks of culture from the original tumor. In this study, the authors demonstrated that both human and mouse tumor organoid-infiltrating T cells exhibited activation, expansion, and cytotoxicity responses to dual PD-1/PD-L1 checkpoint blockade with rapid 7-day assessment. This demonstrated the potential of organoids as tools to predict the clinical response of ICIs therapies. Lastly, a patient-derived tumor fragment model was developed in order to dissect the early immunological responses of human tumor fragment tissue to ex vivo anti-PD-1 therapy [153]. In this platform, fresh biopsies were dissected into fragments and embedded into an artificial extracellular matrix to avoid immune cell efflux. Stable production of cellular and soluble factors was observed up to 48 h of culture, demonstrating preserved TME and architecture. PD-1 blockade response was assessed in 37 tumors from different cancer types (melanoma, non-small lung cancer, breast cancer, ovarian cancer, and renal cell carcinoma). Unsupervised hierarchical clustering analysis was performed on anti-PD-1 treated and untreated conditions. The analysis revealed two groups of tumor distributions: non-responders with minor changes and responders presented an increase in IL-2, TNF-α, and INF-γ secretion compared to non-responders with the release of multiple other cytotoxicity factors. Metastatic colorectal carcinoma fragments were also cultured with the anti-PD-1 pembroluzimab and showed relevant sensitivity to PD-1 blockade therapy [154]. Fragments from human primary breast cancer have been developed to evaluate the effect of different ICIs including anti-PD-1, anti-TIM-3, and anti-PD-L1 [190]. Transcriptomic analyses revealed a unique mechanism of action of these three ICIs, including an upregulation of genes that can enhance anti-tumor immunity and induce tumor suppression/apoptosis to control tumor growth. In the second strategy, tumor organoids were reconstituted with autologous immune cells from peripheral blood to evaluate the therapeutic effect of ICIs [155,191,192]. In this approach, allogenic T and NK cells quickly infiltrated colorectal tumor spheroids, inducing immune-mediated tumor cell apoptosis and destruction. Co-culturing PDOs with autologous PBMCs enables the investigation of responses of endogenous TILs. Indeed, the co-cultures of PDOs generated from mismatch repair-deficient non-small lung carcinoma and colorectal cancer with autologous PBMC induced the expansion of CD8+ tumor-reactive T cells [148]. Yet another study developed rectal cancer organoids co-cultured with patients matched TILs [155]. They evaluated TIL-mediated organoid lysis by measuring cell death for 17 patients. The anti-PD-1 response was also assessed in a subset of patients’ specimens. Results showed that 6 of 17 patients achieved an objective complete response. Assessment of the effectiveness of checkpoint inhibitors revealed a partial restoration of cytotoxicity in TILs with an increased PD-1 expression upon PD-1 blockade. Recently, a matched melanoma/lymph node organoid model was developed [156]. Organoids were screened with various ICIs. Interestingly, the response of these organoids to immunotherapies was frequently similar to specimen clinical response (85%). In addition, peripheral T cells infiltrating the PDO, and subsequently transferred to naïve PDO from the same patient resulted in tumor killing. This result suggested a possible role of immune-enhanced PDOs in generating adaptative immunity. PDOs were also generated from gastric biopsies and resected tumor tissues. Decreased organoid density was observed in response to nivolumab or combinatorial nivolumab plus cabozantinib treatment. Non-small cell lung carcinoma fragments cultured with matched PBMC in the presence of Pembrolizumab revealed extensive tissue damage including the presence of macrophages in apoptotic regions [157]. This response is expected based on Pembrolizumab-mediated activation of exhausted PD-1+ T cells that possess anti-tumor activity.

In an immune-oncology setting, microfluidics-based ‘organ-on-a-chip’ has been used to model ICI in a small number of cancers [193,194,195,196]. This system demonstrated that murine and PDOs were able to retain autologous lymphoid and myeloid cells and responded to anti-PD-1 immune checkpoint blockade in short-term ex vivo cultures [193,194]. Thanks to this, the authors recapitulated features of ex vivo sensitivity and resistance to PD-1 blockade. This study used for the first time a functional assay to evaluate and quantify response to PD-1 blockade in a model containing tumor cells, and autologous stromal and immune cells from explanted tumors [193,194]. Moore et al. used a multiplex microfluidic perfusion system called EVIDENT (Ex vivo Immuno-oncology Dynamic Environment for tumor biopsy) that can accommodate up to 12 separated biopsy fragments that can interact with their matched TILs [197]. This dynamic microenvironment sustained tumor fragments for multiple days. The authors demonstrated that TILs infiltrated organoids and that cytotoxicity was increased in anti-PD-1 treatment conditions. Furthermore, Cui et al. described a patient-derived ‘glioblastoma-on-a-chip’ micro physiological model to dissect the heterogeneity of immunosuppressive TME and assess inhibition of PD-L1 and CSF-1R across various subtypes of glioblastoma [196]. Through this model, they demonstrated that molecular subtypes of glioblastoma have distinct epigenetic and immune signatures that lead to different immunosuppressive mechanisms. Another group co-cultured cholangiocarcinoma organoids with PBMCs or purified T cells [198]. The results showed variable cytotoxicity. When co-cultured with purified CD3+ T cells, the results confirmed that T cells have potent cytotoxic effects on most organoids, suggesting that this strong killing effect of T cells compared to PBMCs was probably caused by higher actual effector cell/target cell ratios. Taken together, these studies demonstrated that the organoid-based propagation of primary tumors with endogenous immune stroma should enable immuno-oncology investigations within the TME with various tumor types and facilitate testing of personalized immunotherapy to predict the efficacy of ICI and elucidate related molecular mechanisms in cancer immunology in vitro.

#### 4.4.2. Monoclonal and Bispecific Antibodies (bsAbs)

In a recent study, organoids were generated from fresh surgical fragments from patients with renal and endometrial carcinoma [159]. Organoids were cultured in the presence of TNF-α and IL-1β, resulting in IL-8 secretion. IL-8 was neutralized when organoids were treated with the TNF-α blockers etanercept and infliximab. A reduction in circulating IL-8 in a plasma sample was also confirmed in a patient who underwent a phase I clinical trial of infliximab monotherapy. NCM460 intestinal stem cell spheroids were generated to explore the molecular mechanisms underlying the inflammation-mediated induction of intestinal tumorigenesis [160]. The treatment of spheroids with TNF-α resulted in an increased cell viability, proliferation, and invasion, even in the presence of 5-FU chemotherapy, suggesting that TNF-α increased chemotherapy resistance. In another study, lung cancer organoids were established from cancer stem cells and co-cultured with HUVECs and MSCs [199]. These organoids formed cohesive cell nests similar to human lung cancer. The authors investigated the effect of an IL-6 blockade on the chemosensitivity of the organoid and suggested that IL-6 could be a novel therapeutic target in lung cancer. Microfluidic tumor organoids-on-a-chip that can model more clinically relevant TME and precisely control the flow of molecules and cells can efficiently introduce functional interactions between cytokines, immune cells, and tumor cells from patients. Three-dimensional human melanoma spheroids were generated to study the synergetic effects of decitabine and interferon-I (IFN-I) [161]. Data demonstrated that the drugs effectively suppressed human melanoma growth and migration in human melanoma spheroids, while apoptosis was augmented, both in vitro and in vivo. The mechanisms of resistance to cibisatamab were analyzed by using seven PDO models from metastatic colorectal cancer [158]. To this end, the authors developed an in vitro co-culture assay with CD8+ T cells to assess cibisatamab efficacy using a bispecific monoclonal antibody that binds to the carcinoembryonic antigen (CEA) expressed by tumor cells and to the CD3 receptor on T cells. Using this platform, they demonstrated a heterogeneous expression of CEA in patient-derived CRC organoids: CEA low were resistant, whereas the CEA high were sensitive to cibisatamab. A large-scale functional screen of dual targeting bsAbs on colorectal cancer organoids was developed [162]. In this study, more than 500 bsABs against various targets were developed and high-content imaging was used to capture the complexity of PDO responses. A bsAB containing an EGFR arm and an LGR5 arm was found to inhibit the largest number of PDOs (52%). A newly developed bispecific and tetravalent antibody targeting EGFR and HER3 called “scDb-Fc” was also developed and tested on primary human colorectal cancer organoid KRAS WT [163]. Suppression of organoid growth was observed, providing strong support for a pan-HER receptor blocking approach to combat anti-EGFR therapy resistance of KRAS WT colorectal cancer tumors mediated by the upregulation of HRG and/or HER2/HER3 signaling. BsAb immunotherapy approaches were also tested in short-term patient-derived high-grade serous ovarian cancer organoids co-cultured with intra-tumoral immune cells to assess the mechanism of action of various types of immune cells [164]. The authors compared the action of a bispecific anti-PD-1/PD-L1 ICI to its monospecific anti-PD-1 and anti-PD-L1 controls. The results demonstrated for the first time that the bispecific antibody uniquely induced the activation of NK cells and most strongly induced a change from CD8 naïve T cells to cytotoxic exhausted progenitors. These changes in both cell types were found to be driven by the down-regulation of the bromodomain-containing protein BRD1. Strikingly, these state changes observed in T, NK cells were recapitulated in vitro and in vitro through the inhibition of BRD1, suggesting that BRD1 inhibitors may have increased efficacy.

#### 4.4.3. Application of Organoids in Tumor Vaccination

Another approach of immunotherapies, called cancer vaccination therapy, is to stimulate immune cells with selected tumor antigens or carcinogenic antigens to induce broad-spectrum immune cell responses, which, in turn, will target the tumor. Many types of cancer vaccine therapies have been tested. They include bacterial vectors, viral vectors, immunogenic peptides, immune cells, dead cancer cells, and oncolytic viruses, all reviewed recently by Hollingsworth [200]. This approach is designed to deliver tumor antigens and adjuvants to activate antigen-presenting cells (APCs), such as dendritic cells (DCs), to initiate and direct immune responses by activating the patient’s adaptative immune system against specific tumor antigens [201]. This will induce the regression of established tumors. A study developed and optimized a formulation of target-specific exosomes loaded with Hiltonol (a TRL3 agonist) to form an in situ dendritic cell vaccine to treat breast cancer [165]. Genetically enriched α-Lactalbumin (α-LA) (a human breast-specific immunodominant protein) on the surface of exosomes was used as a specific tumor-homing protein to enhance targeting capability and immunogenicity. The so-engineered exosomes were shown to accumulate in PDOs from breast cancer, to selectively kill a broad range of cancer cells, and to exhibit minimal toxicity to non-cancer cells. Moreover, they stimulated antigen cross-presentation activity of DCs and induced potent CD8+ T cell responses in vitro. Therefore, the combination of TLR3 agonists with ICD inducers based on cell-free exosomes offers a powerful and novel therapeutic platform for designing DC vaccines for BC.

Organoid culture is also a good tool to discover mutation-associated neoantigens for tumor vaccination [202]. Wang et al. generated hepatobiliary PDOs preserving the neoantigens of the corresponding parental tumors [166]. They selected immunogenic neoantigen-peptides from candidates predicted by multiomics sequencing analysis. By co-culturing candidate peptides with HLA-class I matched PBMCs, they obtained neoantigen peptide reactive T cells that killed tumor organoids and achieved the enhancement of ICI on the T cell-mediated attack at the level of individuals. Using colorectal cancer organoids, another study generated four organoid clones from the same patient that maintained in culture the same exonic mutations [203]. Each organoid clone harbored unique mutations that recapitulated intratumor heterogeneity, suggesting that the clones were not genetically identical. By coupling organoid proteomics and their respective peptide ligandomics, it was demonstrated that tumor-specific ligands from DNA-damage control and tumor suppressor source proteins were prominently presented by tumor cells, coinciding likely with the silencing of such cytoprotective functions. To summarize, this study illustrated the heterogeneity of HLA-peptide presentation in an individual patient and indicated that a multi-peptide vaccination strategy against highly conserved tumor suppressors, for instance, BRCA peptides, might minimize the risk of immune escape. These results are promising and show that in the near future, organoids could be exploited in the research and development of vaccines against tumors. In another study, an optimized protocol was derived for the derivation of cancer stem cell-enriched breast cancer spheres [167]. The expression of glycoprotein gp96 in cancer cells was upregulated by a heat shock protein. gp96 is a type of heat shock protein that plays a vital role in directing and delivering antigens through MHC class I and inducing CD8+ T cell responses. This role of HSPs has been used as a basis for clinical trials to develop anticancer vaccines [204]. Mammosphere viability did not decline after such incubation conditions, suggesting that this optimized protocol allows the development of mammospheres as a potent tool for preparing more immunogenic tumor antigens for use in cancer vaccine production. Recently, a group examined the antitumor effect of intratumoral expression of INF-γ driven by the Semliki Forest virus, an alphaviral vector, in mouse breast cancer spheroids [168]. The results revealed that the infected cells were mostly located on the surface of the spheroids with nonhomogeneous penetration into the inside of the spheroids. The size of spheroids was smaller, indicating the inhibitory effect of the infection. Another group generated ovarian cancer stem cell spheroids that expressed α-gal epitopes [169]. SKOV3 cells were infected with lentivirus to mediate the transfer of α-gal epitopes. The addition of PBMCs was found to significantly induce the cell death of SKOV3- α-gal cells. Interestingly, α-gal-KO mice immunized with SKOV3-α-gal spheroids resulted in extensive production of anti-Gal IgG in serum. Tumor growth was inhibited in mice immunized with α-gal epitopes. Immunized KO mice with SKOV3-α-gal spheroids showed the production of effective antibodies against certain tumor-associated antigens. Mass spectrometry and RNA interference analysis of TAAs revealed that antibodies responding to protein c-erbB-2 may be raised in the sera of mice after immunization with SKOV3-α-gal spheroids. Finally, it was demonstrated that vaccination with SKOV3-α-gal spheroids promoted the production of CD3+CD4+ T cells in vivo. These results suggested that vaccination using cancer stem cell-expressing α-gal epitope could be a novel strategy for the treatment of ovarian cancer. Vaccination against antigens expressed by cancer stem cells with enhanced metastatic potential represents a highly attractive strategy to efficiently target CSCs. Cripto-1 (Cr-1) is an oncofetal protein expressed in the majority of human tumors. Ligtenberg et al. explored the potential of Cr-1 vaccinations to target metastatic melanoma in a preclinical model [170]. They showed that Cr-1 is highly expressed by metastatic B16F10 spheroids. They generated 33 overlapping 15 amino acid Cr-1 peptides able to be recognized by CD8+ T cells using in silico prediction analysis. Subsequently, they evaluated their immunogenicity by testing their ability to be presented and recognized by CD8+ T cells in mice vaccinated with plasmid DNA encoding full-length mouse Cr-1. One of the peptides was able to activate CD8+ T cells to produce INFγ, TNFα, and mCr-16-25 in vitro and in vivo. Vaccination against Cr-1 elicited a protective immune response in mice and resulted in a reduced tumor burden and fewer lung metastases upon subcutaneous challenge with murine B16F10 melanoma spheroids. In a recent study, the same group explored the potential of a Cr-1-encoding DNA vaccine to target CSCs TUBO sphere breast cancer [171]. They observed reduced tumor growth in vaccinated mice after being challenged with a TUBO sphere, suggesting that anti-Cr-1 vaccination holds promise as an immunotherapy for metastatic breast cancer.

PDOs are also used as a predictive tool to study the specificity and cytotoxicity of oncolytic viruses. For example, Raimondi and colleagues generated PDOs from a healthy pancreas and from pancreatic ductal adenocarcinoma to screen the oncolytic adenovirus (OA) response [205,206]. Results demonstrated that OA infected and replicated in cancer organoids. Cancer organoids exhibited different sensitivities to OA, indicating that PDOs could serve as a predictive platform to screen for the selectivity and potency of OA. Recently, a protocol to study the effects of the GFP-measles vaccine virus, and a red vaccinia virus oncolytic virus (OV) was developed in stable breast cancer organoids [172]. The results demonstrated that all oncolytic viruses significantly inhibited cell viability in organoid cultures. The tropism of two commonly used OV, adenovirus (Ad5-Δ24) and reovirus (R124 and jin-3), toward primary gastrointestinal fibroblasts derived from human esophageal, gastric, duodenal, and pancreatic carcinomas was studied [173]. Considerable cell death was observed in the majority of the fibroblast cultures infected with either R124 or jin-3 reoviruses, while Ad5-Δ24 did not induce cell death in the vast majority of fibroblasts tested. They demonstrated that reovirus infection and killing of fibroblasts was mediated by JAM-A expression through the C-terminal PDZ domain. In another study, the specific oncolytic activity of the Zika virus against glioblastoma cerebral organoids was also demonstrated [174]. The authors showed that SOX2 and integrin αvβ5 represent key markers for Zika virus infection in association with suppression of immune response genes. Thus, Zika virus infection provides the possibility for brain tumor therapy.

## 5. Discussion

In summary, we reviewed the development of tumor spheroid and organoid technologies and the important progress made in the establishment of various organoid models in recent years. We also summarized the main applications of organoid models in tumor research, cancer modeling, and treatments. Thus, 3D models are emerging to bridge in vitro 2D cell models and in vivo mice models, gaining popularity for their relevance and their ability to replicate the human tumor-stromal crosstalk [207]. Three-dimensional models can enhance the predictive power and provide a reduction in both financial and time costs during later stages of the drug development timeline, allowing the early detection of ineffective agents, and thus reducing the risk of drug withdrawal from the market. While cancer spheroids can be a useful tool for studying cancer biology and drug development, they have limitations that make them less relevant for personalized medicine compared to cancer organoids, which are derived directly from patient tumors and better mimic the complex microenvironment of the original tumor. Indeed, cancer spheroids are typically generated from cancer cell lines and are generally composed of a single cell type, which may not accurately represent the genetic and molecular heterogeneity of a patient’s tumor. This can limit their usefulness in predicting drug response or developing personalized treatment plans. Second, spheroids lack the complex microenvironment found in patient tumors, including the presence of immune cells, stromal cells, and extracellular matrix components. This can limit their ability to model the interactions between cancer cells and the tumor microenvironment, which can impact the response to different drugs and treatments. Therefore, cancer organoids are a more relevant tool than cancer spheroids for personalized medicine. More generally, organoids are considered a powerful tool for cancer research, and they have successfully been used to study the development and progression of cancer, as well as the response to therapy. Although organoid technology shows great potential in the field of cancer research, there are still many major problems hindering its application. Firstly, although researchers have succeeded in modeling organoids in vitro that have the structure of some organs compared with the original organs in vivo, the structure of these derived organoids is still relatively simple and can only partially reflect the native tissue characteristics. Secondly, many factors in the culture system make it impossible for organoids to fully mimic the physiological function of organs. For example, one of the biggest challenges is the lack of TME, notably the microvasculature, in organoids, which hinders in vivo-like expansion and limits organoid size [208,209,210,211]. How to effectively introduce blood vessels, immune cells, and nerve cells into the culture system is also a major problem for studying the influence of TME on cancer behavior against immunotherapy agents. To overcome these restrictions, organoids have been co-cultured with immune cells (PBMC) from lymph nodes, including T cells to model the priming/activation, T cells infiltration inside the tumor, and finally the recognition and killing of cancer cells by effector T cells [212]. The current microfluidic platform used to establish vascularized organoids is semi-adjustable and is affected by numerous factors including flow rate and the concentration and composition of the cytokines [213]. More accurate and flexibly controllable microfluidics platforms are needed for a better vascularization of organoids and an accurate prediction of antiangiogenetic therapies. In addition, long-term maintenance of immune cells in organoid cultures is still hard to achieve. For the long-term preservation of immune cell cultures, using anti-CD3, anti-CD28, and IL-2 cytokine has been suggested [148]. Numerous clinical trials are underway to evaluate the various applications of organoids and their effectiveness in precision cancer immunotherapy (https://clinicaltrials.gov/) accessed on 14 March 2023. The culture media composition for immune cells should be optimized in such a way that can support the growth and function of cells. In addition, technologies have been developed (and are continuing to be developed) that improve long-term culture conditions and the delivery of nutrients and gaseous exchange to the developing organoid [213]. These include spinner flasks and bioreactors to increase “flow” in the culture system and microfluidics-based platforms for efficient nutrient diffusion, oxygenation, and waste metabolite disposal. It is also interesting to see the evolution of permeable membranes such as the Transwell^®^ (Merckmillipore, 0.4 µm) permeable supports and other semi-permeable membrane materials being integrated into perfusion systems and 3D bioprinting techniques to improve nourishment to the organoid during maturation [64,121]. These technologies have helped to increase the lifespan and utility of organoids to months. Third, the large variabilities between 3D models limit their level of standardization and reproducibility, suppressing their use as preclinical tools for drug development. Batch-to-batch variation, cellular constitution, and architecture of organoids are also possible factors affecting reproducibility. A standard procedure for organoid culture methods that supports the efficient generation of organoids from different cancer types is needed to minimize this variability and facilitate their application in high-throughput drug screening, where biopsy samples used for the generation of organoids represent a smaller part of the original tumor. Fourth, the higher heterogeneity of tumors questions the reliability of the substitution of small fragments for the whole tumor. Tissue samples extracted from different sites of the original tumor might better reflect this heterogeneity and facilitate cancer translational research. Fifth, from another point of view, there are also some ethical concerns. For example, the use of embryonic stem cells to generate organoids raises ethical concerns. Sixth, organoids generated from patient-derived biopsies may contain genetic or other sensitive information about the patient, and researchers must take appropriate steps to protect the patient’s privacy. Personalized medicine enables targeted treatment for an individual at a molecular and pharmacogenomic level to maximize the effects of treatment. Organoids, derived from an individual’s stem cells, progenitor cells, or from induced pluripotent stem cells, can be used for disease modeling to test the efficiency and dosage of a drug, and for regenerative medicine, all at a personalized level. Due to their unique ability for unlimited self-renewal, organoids can be exposed to varying drugs to identify the best treatment to fight that particular cancer; thus, personalizing medicine to treat disease. Taking this idea even further is the ability to repair genes in cells that can form organoids, then using those organoids to understand treatment regimens. However, personalized medicine and drug screening applications still need to overcome challenges such as drug delivery. Indeed, organoids are typically grown in vitro, and it can be difficult to deliver drugs to the organoids in the same way as that in vivo. Additionally, organoids may not respond to drugs in the same way that tumors in vivo do, which can limit their utility as a tool for drug development. Another challenge in high-throughput pharmacological and toxicity screening applications has been the formation of reproducible, single organoids per well. The ULA surface of microplates coupled with established biological hydrogels has provided a platform to generate uniformly sized organoids compatible with HTS applications. Concurrent advancements in high-content screening platforms have also helped to elucidate the 3D complexity of organoids in terms of multi-parameter imaging and quantitative analysis. Nevertheless, results obtained from organoids generated from different patients should be validated against other types of experiments, such as animal models, to be able to draw sound conclusions.

## 6. Conclusions

Tumor organoids mimic the primary tumor tissues in both architecture and function, and retain the histopathological features, genetic profile, mutational landscape, and even responses to therapy of the primary malignant cells. Here, we review the current use of tumor organoids as a tool for basic and translational research. Taken together, it is very tempting to implant organoid platforms in clinical practice; however, significant issues still have to be resolved. One major bottleneck that needs to be overcome for potential clinical applications is the development of standardized and robust organoid assays with predefined cutoff values for drug response assessment.

While we are still facing many obstacles on the road to organoid-based clinical decision making, we foresee great potential for the use of organoids in preclinical research, drug screening, immunotherapy prediction, and personalized medicine. As a research tool, organoids also provide unique opportunities for omics disciplines.

In the future, coupling 3D models with high-throughput screening methods, high-content imaging approaches, and proteomics and bioinformatics tools will allow these models to become fundamental tools in pharmaceutical development and biomedical research, especially in the field of immunotherapy. In order to help scientists to adapt their work from monolayers to 3D cell cultures, we believe that a systematic description of 3D culture and analysis methods, as described in this review, will be of great benefit.

## Figures and Tables

**Figure 1 cells-12-01001-f001:**
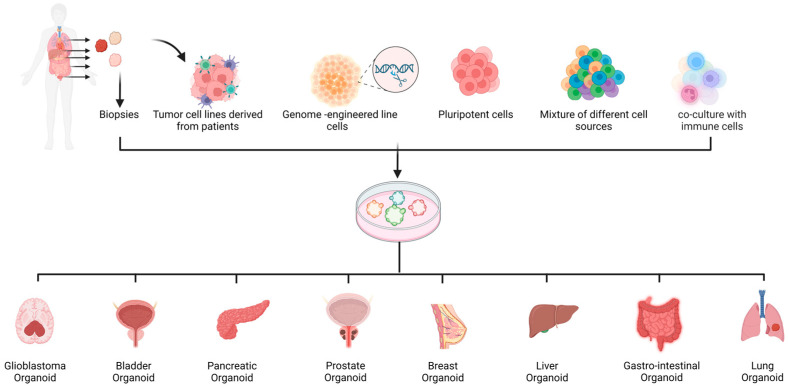
Summary of the procedures used to establish cancer spheroids and organoids. Patients’ primary cells, CRISPR/Cas9 engineered cells, pluripotent stem cells, and a mixture of different cell types and co-culture with immune cells can be used to establish organoids. This figure was created with BioRender.com (accessed on 14 March 2023).

**Figure 2 cells-12-01001-f002:**
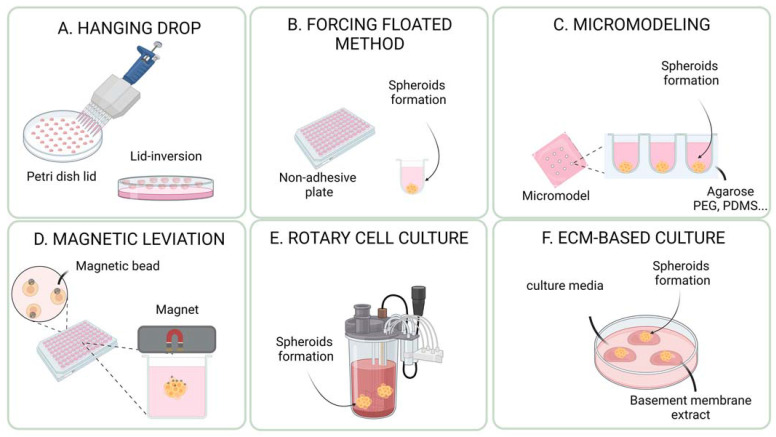
Schematic representation of the different 3D cell culture techniques: (**A**) In the hanging method, the cells form a single spheroid by accumulating at the free liquid interface formed by their suspension due to the inversion of the dish. (**B**) The forced floating method can be carried out using uncoated polystyrene plates or plates coated with a hydrophilic polymer that suppresses cell-substrate interactions, e.g., ultra-low attachment (ULA) plates. (**C**) In the modeling culture system, the cells are seeded and allowed to self-aggregate into non-adhesive micro-molds. (**D**) The magnetic levitation employed magnetic beads assembly technique. Cells treated with magnetic beads aggregate into spheroids or organoids under magnetic forces within a few hours after a magnet is placed on top of the lid. (**E**) The Rotary Cell Culture System is another agitation-based technique used to obtain a higher number of large spheroids with a small number of starting cells. (**F**) Organoids can be cultured in the submerged method by disrupting tissue mechanically and enzymatically into single-cell suspensions, followed by embedding them in basement membrane extract (BME) and submerging them in the culture media. This figure was created with BioRender.com (accessed on 14 March 2023).

**Figure 3 cells-12-01001-f003:**
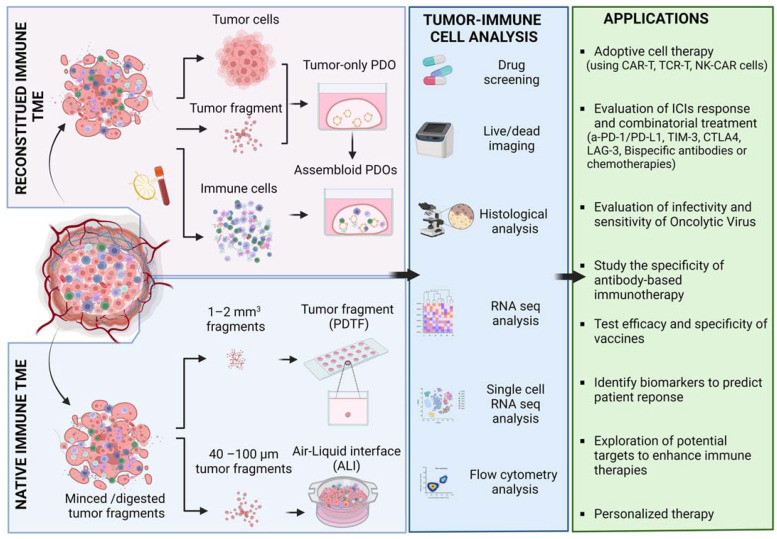
Application of organoids in Cancer Immunotherapy. The TME can be generated in PDOs by two types of approaches: the reconstituted TME (Upper panel, pink color) and the native TME (Below panel, blue color). In the reconstituted system, PDOs were exclusively generated from tumor cells after physical or enzymatical digestion. Organoids are cultured in extracellular matrix dome (Matrigel or BME (Basement Membrane Extract)). Organoids were co-cultured with autologous PBMC isolated from peripheral blood, lymph node, or from tumor. In native TME model, tumors were minced or enzymatically digested, then filtered to obtain appropriately sized fragments, embedded into collagen-based extracellular matrix with medium in the top. Alternatively, in air–liquid interface (ALI) culture, native tumor fragments are embedded in collagen matrix on the top of a transwell insert exposed to air with culture medium below it. Numerous downstream applications of organoids in immunotherapy followed by systemic functional and quantitative analysis are summarized in the right panel. It allows the interaction between immune cells and tumor cells to be defined, in order to identify and predict the clinically relevant immunotherapy strategy for patients. Abbreviations: CAR, chimeric antigen receptor; NK, Natural killer; ICI, immune checkpoint inhibitors, CTLA4, cytotoxic T-lymphocyte associated protein4; PD-1, programmed cell death-1; LAG-3, Lymphocyte-activation gene 3; TIM-3, T-cell immunoglobulin and mucin containing protein-3. This figure was created with BioRender.com (accessed on 14 March 2023).

**Table 1 cells-12-01001-t001:** Overview of the current sources to generate and culture in vitro cancer spheroids and organoids; advantages and inconvenience.

Cell Sources	Advantages	Inconvenience
Cancer cell lines	Readily available and can be purchased from various sources Low cost Easy to cultivate and maintain Can be used in high-throughput screens to identify new therapeutic targets and drug candidates	Not representative of tumor heterogeneity May not represent the in vivo microenvironment Can acquire genetic and epigenetic changes over time and may not reflect the original tumoral cell
Primary cells	Closer representation of the in vivo microenvironment of the patient’s tumor. More accurate reflection of the genetic and molecular changes that occur in the patient’s tumor Useful to test the effects of drugs and other therapies on the patient’s tumor (personalized medicine)	Difficult to obtain, especially for certain types of tumors Challenging to culture and maintain in the lab, especially for certain types of cells May not be representative of all patients, since each patient’s tumor is unique
Genetically modified cells	Study of specific genetic pathways or specific mutations that are found in a patient’s tumor	Not replicate the complexity of the in vivo microenvironment Not representative of tumor heterogeneity Not representative of the genetic diversity of the tumoral cells
Pluripotent stem cells	Organoids generation by differentiating them into cells that are similar to those found in a patient’s tumor Can be used to study the effects of genetic changes on tumor development Can provide a large number of cells for research	Technically challenging to differentiate into the specific cell types found in a patient’s tumor Not fully replicate the complexity of the in vivo microenvironment Not representative of tumor heterogeneity
Mixture of different cell sources	Better representation of the tumor microenvironment Study of the interactions between different cell types in the microenvironment Provide insights into the mechanisms of tumor progression and response to therapy. Can be used in high-throughput screens to identify new therapeutic targets and drug candidates	More difficult and complex to culture and study More difficult to scale up than those made from a single cell type, which can limit their use in high-throughput screens and other applications Not fully represent the complexity of the tumor microenvironment
Organoids including immune cells	Closely mimic the complex interactions between cancer cells and the immune system that occur in vivo Can provide important insights into the mechanisms of immune response and resistance Can be used in personalized immunotherapy applications Can be used to identify new therapeutic targets and drug candidates that modulate the immune response	It is challenging to include and maintain immune cells that are present in vivo More complex to culture and study More difficult to scale up than those made from a single cell type, which can limit their use in high-throughput screens and other applications

**Table 2 cells-12-01001-t002:** State of the art of methods for the generation and culture of cancer spheroids and organoids: advantages and inconvenient.

Method or Technology	Advantages	Inconvenient
Hanging drop method	Simple and easy to perform Low cost Generate small numbers of organoids No need for specialized equipment	Can be technically challenging to handle and manipulate the organoids Limited scalability and reproducibility Takes a long time Difficult long-term culture
Low-attachment plate method	Can be used to generate small or large numbers of organoids. Simple and easy to perform Wide range of cell types and seeding concentrations	Formation of spheroids with irregular and heterogeneous shapes and sizes Not provide adequate oxygen and nutrient supply to the center of the spheroids Cost of the equipment Continuous constant agitation could be a need to culture spheroids and organoids Challenging medium exchanges
Microwell method	Allows for the formation of multiple organoids in a single well Allows for high-throughput screening Allows for better control over the microenvironment of the organoids Allows for the formation of multiple organoids in a single well	Can be technically challenging to handle and manipulate the organoids High cost of the equipment Can be technically challenging to handle and manipulate the organoids Relatively high cost of the equipment
Magnetic levitation	Formation of uniform cell aggregates Can be used to create dynamic and versatile microenvironments for cells, (ex: mimicking the mechanical forces of blood flow) Using magnetic fields to manipulate the position of cells in real time	New technology and more research are needed to fully understand its potential uses in cancer research High cost of the specialized equipment and the magnetic beads Difficult to scale up to larger numbers of cells or larger cell cultures Cell type compatibility of magnetic beads May require additional steps to remove the beads after the cells have formed spheroids or organoids
Rotatory systems (Bioreactors and spinner flask)	Can generate organoids in a controlled and reproducible manner Mimic the in vivo microenvironment of tumors Can culture large numbers of organoids in a small volume of medium	Complexity of the technology High cost of the equipment Need of very specialized equipment Can be difficult to handle and manipulate the organoids High shear stress and for cells
Extracellular-matrix-based method	Can mimic the in vivo microenvironment of tumors Mimicking cell-to-ECM interactions Improvement of the TME supportive and physiologically relevant environment for the cells to grow and differentiate	Can be technically challenging to handle and manipulate the organoids High cost of ECM products Extensive batch-to-batch variability Poor control of mechanical properties Choice of the appropriate ECM

**Table 3 cells-12-01001-t003:** Overview of possibilities of tumor organoids in immunotherapy research.

Application	Cell Source	Culture System	Cancer Type	References
Chemotherapy	Primary biopsy	Embedded or resuspended in Matrigel, BME Type 2, ECM-mimicking HA/collagen-based Hydrogel	Colorectal, pancreatic cancer, metastatic colorectal cancer, breast cancer, ovarian cancer and appendiceal cancer	[35,62,63,119,142,143,144]
CRISPR/Cas9 therapy	HEK293T OVCAR5 cells HCT116 cells iPSCs Human and mice biopsies	Embedded in Matrigel, ultra- low-attachment 6 well culture plates, Drops of Matrigel	Mice tubo-ovarian carcinoma, ovarian cancer cells, colon cancer, murine and human small intestine and pancreatic ductal adenocarcinoma	[70,84,85,86,145,146,147]
Adoptive cell transfer therapy	Tumors tissues Biobank PDOs SKOV3 human epithelial ovarian cancer cells biopsies	Embedded in Geltrex Basement membrane extract (BME, Cultrex), PDMS fluidic chamber, ultra-low attachment 6-well culture plates, Matrigel, agar-coated plates, embedded in growth factor reduced Matrigel	Colorectal cancer, non-small cancer lung cells, breast cancer and normal breast, Human ovarian cancer cells, glioblastoma and bladder cancer	[57,148,149,150,151,152]
Immune checkpoint inhibitors	Biopsies	ALI, Special cell-culture inserts, embedding in Geltrex, BME, multiwell plates, embedded in 3% low-melting agarose	Melanoma, colorectal cancer, non-small cancer lung cells, breast cancer, ovarian cancer or renal cell carcinoma, colorectal carcinoma, gastric cancer and lung cancer	[64,148,153,154,155,156,157]
Monoclonal and bispecific antibodies	Biopsies, colon epithelial cell line NCM460, A375 human melanoma cells, Prospect C and Prospect R trials	Resuspended in 100% Matrigel, ultra-low attachment culture dishes, embedded in growth factor reduced Matrigel	Human colorectal cancer, colon cancer line, melanoma cancer and metastatic colorectal cancers	[158,159,160,161,162,163,164]
Cancer vaccines	MDA-MB-231, MCF7, MDA-MB-435S, SKBR3 cancer cells, 4T1/eGFP cells, SKOV3 cells, B16F10 cells TUBO cells, 4T1 cells, HT29 colorectal cancer cells, biopsies	Embedded in 4% agarose, BME; cultrex PC BME RGF type 2, 24-well plates a thin layer of 1.5% agarose, ultralow attachment plates, cancer cells grow as non-adherent spheroid cells, ultra-low attachment plates, Matrigel	Breast cancer, hepatobiliary tumor, colorectal cancers, ovarian cancer, metastatic melanoma, murine mammary carcinoma and glioblastoma.	[165,166,167,168,169,170,171,172,173,174]

## Data Availability

Not applicable.

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
