# Peer review of "Cancer Spheroids and Organoids as Novel Tools for Research and Therapy: State of the Art and Challenges to Guide Precision Medicine"

_cells, 2023, doi:10.3390/cells12071001_

Round 1

Reviewer 1 Report

General comments

The manuscript reviews a hot topic in cell culture/drug discovery. The manuscript describes the main sources and methods to generate cancer spheroids and organoids, the main applications of organoids models in tumor research, cancer modeling and possible treatments.

Although there are already several reviews in this area, still they focus on different aspects and different points of view which I believe can be useful for investigators in this area. Generally, the manuscript is very complete and successfully written. However, I suggest several points of improvement.

There are several formatting problems in the manuscript, examples of this are missing some spaces after full points e.g line 41, or missing the final stop in line 65. Different sizes of letters, different spaces among paragraphs and others. Please revise all these minor formatting topics in the manuscript.

Although it is an extensive manuscript, which is normal as it is a review, there are several paragraphs without references. Most are made as an introduction to the subject, however even being so, relevant references, for instance, reviews on that specific topic could have been included.

1.      Line 36. Suggest better mimicking, as 3D cell cultures do not fully represent the in vivo

2.      From lines 32-65: There is no reference, please insert appropriate references in some key sentences such as after each item of “(i) understanding the pathophysiology of cancer progression and resistance [REF], (ii) in vitro screening of anti-cancer treatments [REF], and (iii) reproducing in vitro the specificity of one patient’s tumor, to allow a personalized screening of the most effective treatment [REF]. or “Spheroids can be generated from a wide variety of cell types, including primary cells, cancer cell lines, and cancer stem cells.” [REF]. and so forth ...along all sections 1. Introduction

3.      From my point of view, point, 2. is also an Introduction, consider giving a title like 1.1 3D cell culture – spheroids and organoids for the piece of text from lines 32-65, and the next topic could be 1.2 Current sources to generate and culture in vitro tumor spheroids and organoids.

4.      Please include in line 53 what CAR-T stands for.

5.      Line 57-58: There is something wrong with the sentence” To try to solve these limitations, the addition of non-autologous stromal/immune cells to 3D culture is unfortunately not optimal”, can you check, please? Is not yet optimized?

6.      Line 60: the development

7.      Line 72: Star point 2.1 with Capital Letter

8.       Lines 74-76: Introduce appropriate references.

9.      Line 81: Add the information that the referred study was performed using pancreatic cancer cell lines, the authors can also refer to the name of the cell lines.

10.  Lines 105-111: Please insert some appropriate references for the limitations of the use of cancer cell lines for spheroids generation.

11.  Line 136: Remove one full stop

12.   Line 158: Reference 45 does not correspond to Norman Sachs et al 2018.

13.  Lines 160-164: Are there any appropriate references for the pointed disadvantages of biobanks?

14.  Line 179: Please introduce what CRISPR/Cas9 stands for.

15.  Line 189: Please introduce what AKSTP stands for.

16.  Line 192: Please introduce what AKPS stands for.

17.  Line 239: Please insert an appropriate reference for these first sentences.

18.  Lines 290-292: Please insert an appropriate reference for these first sentences.

19.  Lines 315-310: Please insert some appropriate references for the referred limitations.

20.  Lines 345-351: Please insert some appropriate references for the referred limitations.

21.  Line 353: Please refer to Figure one in the text, in a place that makes sense, before the image.

22.  Line 361: There is no reference in all section 3.1  Add appropriate ones

23.  Line 361: please write ECM (extracellular matrix), as it is the first time it appeared

24.  Lines 422-427: Please insert appropriate references to the pointed ULA plates characteristics

25.  Lines 442-448: Please insert appropriate references to the pointed ULA plate limitations.

26.  Lines 451-456: Please insert appropriate references to the pointed microwells technique description

27.   Lines 490-493: Please insert appropriate references to the description of the magnetic levitation technique

28.  Lines 525-532: Please insert appropriate references to the advantages of bioreactors

29.  Line 555: Please insert what MCTS stands for

30.  Line 561: If the authors refer to several limitations, several limitations should be enumerated and supported by appropriate references

31.  Figure 2: letter E Rotary cell culture or rotatory cell culture is wrong in the figure

32.  Tables 1 and 2 can be presented without references as long as in the text all the missing references suggested above are presented.

33.  Lines 713-721: Add appropriate references

34.  Line 753: Please insert what ICIs stand for

35.  Figure 3: the letters of the first two topicas are a little bit difficult to read, I would suggest increasing the size of the letter, also uniformized the last topic initiating with a capital letter

36.  Line 950: in PDO? Instead of is PDO

37.  Line 955: native

38.  Table 3 is completely misaligned, it is very difficult to follow which information corresponds to the next collum and corresponds to which reference.  As the table occupies more than one page the head of the table should be repeated on each page. See for instance table 1 of a recently published review in Cells journal Cells 2023, 12(3), 429; https://doi.org/10.3390/cells12030429

39.  Table 3: Write ICIs not I acronym

40.  The Discussion lacks also lacks references to sustain the presented facts. The discussion should also refer to something relative to the use of organoids in Immunotherapy, as it is a strong point focused on the manuscript.

Author Response

We would like to thank the reviever for his/her comments, and also for his/her time and effort that he/she has put into evaluating our manuscript. We hope that you will be satisfied by the corrections.

Reviewer 2 Report

The authors summarized the characteristics of spheroids and organoids in drug development and described the usefulness of organoid in personalized medicine. This review manuscript is well written.

In the Discussion and Conclusion section, organoids were mentioned as a powerful tool in translational research because organoids could be generated from same patient-derived tissues. However, spheroids were not described in detail in these sections. Because spheroids are composed of a single cell type, they are difficult to use as a tool in personalized medicine. The authors should add this point.

Author Response

We would like to think the reviewer for his/her positives remarks and for his/her effort that he/she putted for evaluating our manuscript. We hope we have answer to all the comments.

Reviewer 3 Report

The manuscript of El-Harane et al. highlights the most relevant work about the use of PDO for screening the efficacy of therapeutics. The manuscript Is well written and clear. And I recommend it to be publish. There are only few minor suggestions. 

1. Authors when describing the use of Matrigel to mimic the ECM, are missing the use of other biomaterials that are recently been used to recapitulate the ECM. They just mention differences among Mariel and gelatin, alginate and collagen in cell lines. Therefore, it should be also address.

2. There are some typos in the text, please fix them. For instance, some words such as in vitro and in vivo should be in italic

3. check the reference 171

Author Response

We thank the reviewer for its positive comments on our work. We do agree with his/her comments. We hope that he/she will be satisfied by the changes we have made in the manuscript.

Round 2

Reviewer 1 Report

The authors performed most suggested alterations. The manuscript can now be accepted.